# SUPERVISED FINE-TUNING ON AMBIGUOUS PREFERENCE PAIRS BOOSTS LLM ALIGNMENT

## ABSTRACT

Preference learning constitutes a fundamental component in aligning large language models (LLMs) with human values and ethical expectations, where the quality of preference data plays a critical role. Existing methods typically assess data quality by measuring the margin between preferred and dispreferred responses in each pair. Following the common intuition that small-margin (i.e., difficult) pairs are uninformative or even noisy, such pairs are often discarded. In this work, we challenge this natural practice and propose a new insight: *"Despite difficult pairs may hinder alignment when optimized with preference-based objectives due to potential likelihood displacement, they can still provide valuable learning signals when trained with supervised fine-tuning (SFT)."* We empirically validate this insight through systematic experiments and highlight two key findings: (1) Structuring training from easy to difficult samples improves alignment performance, consistent with the curriculum learning paradigm; (2) Difficult pairs negatively impact preference-based optimization but become useful when optimized using SFT loss. Based on this insight, we introduce a simple yet effective method, **MixDPO**, which ranks preference pairs by difficulty and dynamically switches to SFT loss for difficult pairs. Our approach achieves improved alignment performance on the AlpacaEval 2 benchmark, outperforming existing DPO variants, particularly for the Length Control (LC) win rate.

## 1 INTRODUCTION

Learning from human feedback is essential for aligning large language models (LLMs) with human preferences, helping ensure that these models behave in ways that are helpful, honest, and harmless (Achiam et al., 2023; Nakano et al., 2021). A widely adopted approach for such alignment is reinforcement learning from human feedback (RLHF) (Stiennon et al., 2020; Ouyang et al., 2022), which involves a multi-stage pipeline of LLM fine-tuning and reward model training. To simplify this complex process, several off-policy and reward model-free approaches have been proposed, including Direct Preference Optimization (DPO) (Rafailov et al., 2024b) and its variants like KTO (Ethayarajh et al., 2024) and SimPO (Meng et al., 2024), to name a few. These approaches bypass online reinforcement learning by directly training on a fixed dataset of preference pairs $\{(x, y_w, y_l)\}$, where $y_w$ and $y_l$ represent the preferred and dispreferred responses given the prompt $x$, respectively. Unlike the open-ended exploration used in RLHF, these methods rely heavily on the quality of the underlying preference data, which is crucial for achieving strong alignment performance.

Despite considerable efforts having been devoted to curating post-training data, either at the sample level (Chen et al., 2023b; Xia et al., 2024; Pang et al., 2024a; Liu et al., 2023) or the token level (Lin et al., 2024; Pang et al., 2025), the role of data in preference alignment remains heavily overlooked and underexplored. Building on the idea of data selection, recent work (Deng et al., 2025; Huang et al., 2025; Gao et al., 2025) investigates pairwise data quality based on the margin between preferred and dispreferred responses within each pair, aiming to identify easy, high-margin pairs. In contrast, those *difficult* (low-margin) pairs are often discarded based on the intuition that they introduce ambiguity or noise, hindering effective preference modeling (Deng et al., 2025; Gao et al., 2025).

This rationale stems from the observation that such difficult pairs often induce a *likelihood displacement* phenomenon (Pal et al., 2024; Yuan et al., 2024; Rafailov et al., 2024a; Razin et al., 2024), where the log probability of both the preferred response $y_w$ and dispreferred response $y_l$ decreases,

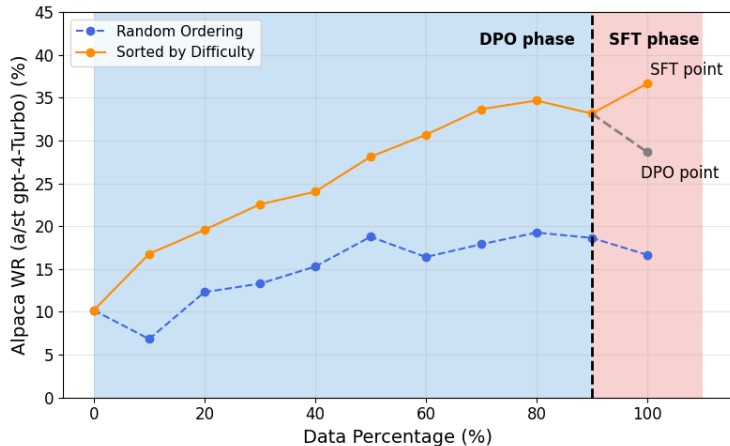

Figure 1: Performance comparison between random ordering and difficulty-based sorting on the AlpacaEval 2 benchmark. DPO is used as the default loss function. Notably, instead of discarding difficult preference pairs, further training them using the SFT loss leads to improved performance.

contradicting the initial goal of preference learning. The potential reason is that subtle differences between similar responses can lead to unstable gradients and misaligned optimization. Most existing approaches attempt to alleviate this issue by redesigning DPO-style loss functions, such as Cal-DPO (Xiao et al., 2024) and DPOP (Pal et al., 2024). In contrast, from a data-centric perspective, we raise an interesting question that *"Can these difficult pairs, which are typically discarded, still offer valuable supervision signals if properly utilized?"*

In this work, we challenge the conventional practice by proposing a new insight that *"Despite difficult pairs may hinder alignment when optimized with preference-based objectives due to potential likelihood displacement, they can still provide valuable learning signals when trained with supervised fine-tuning (SFT)."*

Our empirical experiments in Figure 1 support two key findings: (1) Given the natural variation in difficulty across preference pairs, structuring training from easy to difficult pairs, compared with random ordering, leads to better alignment performance, consistent with the curriculum learning paradigm; (2) While difficult pairs often harm preference-based optimization, they become beneficial when trained using SFT loss. Ultimately, backed by empirical evidence, we introduce a simple yet effective method that adaptively switches from DPO to standard SFT loss based on the difficulty of preference pairs. Specifically, we apply DPO loss to optimize easier pairs, while resorting to SFT loss for the more challenging ones to better harness their informative value. Importantly, SFT loss provides a straightforward way to avoid likelihood displacement.

We summarize our main contributions as follows.

- We conduct a systematic study on the role of easy and difficult preference pairs in alignment performance, revealing a consistent pattern: performance deteriorates as training data shifts from easy to difficult pairs.

- Challenging the common practice of discarding difficult pairs with small score margins, we propose a novel insight—these pairs can still offer valuable learning signals when optimized using supervised fine-tuning (SFT), as confirmed by empirical evidence. Motivated by this, we introduce a simple yet effective method, **MixDPO**, which applies DPO loss to easy pairs and SFT loss to difficult ones to better utilize informative signals.

- We perform extensive experiments to validate the effectiveness of **MixDPO**, against DPO and several widely-used variants, including CPO, IPO, KTO, SimPO, and SelectiveDPO. Notably, the introduced SFT phrase serves to counteract the tendency toward overly long responses driven by preference bias, as evidenced by improvements in the AlpacaEval 2 LC win rate. The extensive ablation study on the additional base model and Agrilla-7k dataset demonstrates the generality and adaptability of our approach.

## 2 RELATED WORK

**LLM Preference Alignment** Start from work (Ouyang et al., 2022), numerous approaches have been proposed to align LLM-generated responses with human preferences. These methods can be broadly categorized into two paradigms: Reinforcement Learning from Human Feedback (RLHF) and Direct Preference Optimization (DPO) (Rafailov et al., 2024b). As a simplified, reward model-free alternative to RLHF, DPO has emerged as one of the most widely adopted techniques for preference alignment. A growing body of research has focused on analyzing the DPO loss from various theoretical and practical perspectives, leading to several notable variants, including IPO (Azar et al., 2024), KTO (Ethayarajh et al., 2024), CPO (Xu et al., 2024a), and SimPO (Meng et al., 2024), to name a few.

While substantial effort has been devoted to data selection in instruction tuning and pre-training at both the sample level (Chen et al., 2023b; Xia et al., 2024; Pang et al., 2024a; Liu et al., 2023) and the token level (Lin et al., 2024; Pang et al., 2025), the data impact on the preference alignment remains overlooked and underexplored. Several prior studies (Bai et al., 2022; Ethayarajh et al.) have examined the influence of data during alignment, but primarily from the perspective of aligning models with human values and ethics. Several recent work (Deng et al., 2025; Huang et al., 2025; Gao et al., 2025) investigates data quality based on different metrics such as reward score, aiming to retain informative samples while discarding those deemed uninformative or noisy. Another line of work focuses on developing noise-tolerant DPO objectives, such as cDPO (Mitchell), robustDPO (Chowdhury et al., 2024), and PerpCorrect (Kong et al.), but does not investigate the impact of individual samples. In contrast, our work challenges this binary view of data quality. We argue that seemingly uninformative or overly difficult samples that are typically filtered out in preference-based optimization, can still provide valuable information when optimized using supervised fine-tuning (SFT) objectives.

**Curriculum Learning** Curriculum learning (CL) follows the natural human learning patterns by structuring learning from simpler to more complex concepts (Avrahami et al., 1997; Bengio et al., 2009), which could effectively accelerate model convergence and enhance generalization. Inspired by its success, CL patterns have been incorporated into several domains, including machine translation (Platanios et al., 2019), image generation (Croitoru et al., 2024b), and multimedia search (Jiang et al., 2014; Tudor Ionescu et al., 2016).

In preference alignment for LLMs, a central component of curriculum learning is the definition of sample difficulty on preference pairs. Recent studies have explored various difficulty scoring metrics to enable curriculum-based training, such as prompt length or inherent attention scores (Kim & Lee, 2024), model perplexity on responses (Wu et al., 2024), reward margins estimated by strong reward models (Croitoru et al., 2024a), and validation DPO loss on preference pairs (Gao et al., 2025). In this work, we adopt a simple yet effective strategy by using the original rating scores available in the preference data to estimate difficulty, incurring no additional computational overhead. Moreover, while (Gao et al., 2025) opts to filter out overly difficult examples due to their potential negative impacts, we present an alternative approach, demonstrating that these samples can still contribute to alignment when optimized using their SFT loss.

## 3 PRELIMINARY

### 3.1 DIRECT PREFERENCE OPTIMIZATION (DPO)

Preference alignment (Ouyang et al., 2022) aims to ensure that large language models (LLMs) generate outputs that reflect human preferences and communication styles, thereby enhancing their safety, reliability, and trustworthiness in real-world settings. In this work, we adopt Direct Preference Optimization (DPO) (Rafailov et al., 2024b), one of the most widely used methods for preference-based alignment. Given a pairwise preference dataset $\mathcal{D} := (x, y_w, y_l)$, where $x$ is a prompt, $y_w$ is the preferred response, and $y_l$ is the less preferred response, DPO trains a policy model $\pi_\theta$ using the following objective:

$$\mathcal{L}_{\text{DPO}}(\pi_\theta; \pi_{\text{ref}}, \mathcal{D}) = -\mathbb{E}_{(x,y_w,y_l)\sim\mathcal{D}} \left[ \log \sigma \left( \beta \log \frac{\pi_\theta(y_w \mid x)}{\pi_{\text{ref}}(y_w \mid x)} - \beta \log \frac{\pi_\theta(y_l \mid x)}{\pi_{\text{ref}}(y_l \mid x)} \right) \right], \quad (1)$$

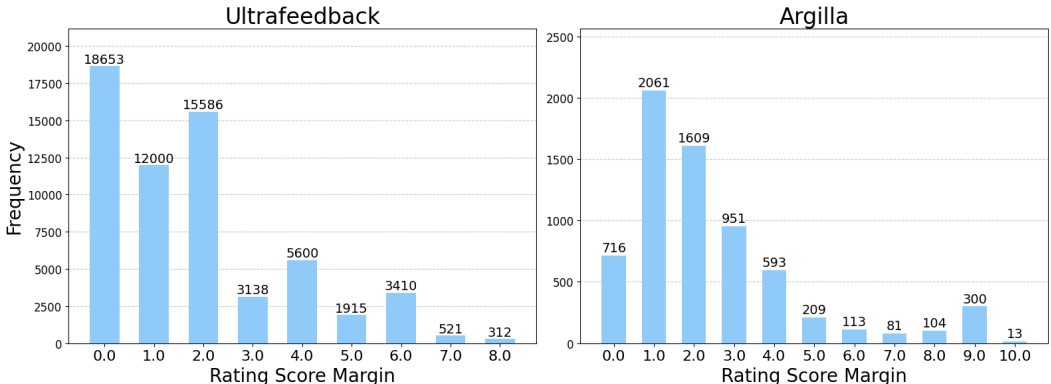

Figure 2: Original rating score margin distribution. **Left**: Ultrafeedback dataset (61k samples). **Right**: Argilla dataset (7k samples). Chosen/Rejected scores are both annotated from various LLMs. Observe that approximately 50% of the samples in both datasets exhibit a score difference below 1.0.

where $\pi_{\text{ref}}$ represents the reference policy model—typically the model fine-tuned via SFT, $\sigma(\cdot)$ denotes the sigmoid function, and $\beta$ is a hyperparameter that controls the divergence between the reference model $\pi_{\text{ref}}$ and the current policy $\pi_\theta$.

## 3.2 CURRICULUM LEARNING

Curriculum learning (Bengio et al., 2009) is a training strategy that gradually exposes the model to increasingly difficult examples, inspired by how humans learn. Curriculum learning can improve convergence speed and model generalization by starting with easier samples and progressively moving to harder ones. This paradigm relies on assessing sample difficulty to guide the model from learning on easier examples to more challenging ones over time.

**Qualifying pairwise difficulty** Note that preference datasets used for DPO are typically constructed from response pairs annotated with LLM-generated rating scores. In practice, for each prompt, the response with the higher rating score is selected as the chosen response, while the one with the lower score is designated as rejected. Building upon this construction, the rating scores can naturally serve as a basis for estimating the sample-level difficulty. We define this notion as follows:

**Definition 3.1** *(Pairwise Difficulty) Given a preference pair represented as $(x, y_w, y_l, s_w, s_l)$, where $s_w$ and $s_l$ denote the rating scores of the chosen and rejected responses, respectively, the pairwise difficulty is defined as the rating score difference:*

$$M(s_w, s_l) := s_w - s_l \tag{2}$$

In practice, metric $M(\cdot)$ is always non-negative. By default, the rating score margin ($M$) will be utilized to measure pairwise difficulty, where a higher $M$ implies an easier pair. Intuitively, preference pairs within a dataset can differ in difficulty, as reflected by the varying rating score margins between the chosen and rejected responses. Figure 2 presents the score gap distributions of two popular preference datasets, including Ultrafeedback and Argilla, highlighting the widespread presence of difficult pairs in the dataset.

While Definition 3.1 serves as our primary metric, alternative metrics can also be considered, such as reward score margin (Croitoru et al., 2024a), DPO loss (Gao et al., 2025), or the embedding distance between responses. Note that our goal is not to develop a superior metric for difficulty estimation. Instead, we investigate the role of difficulty based on the available annotations used during dataset construction. Notably, rating scores are computation-free, whereas model-dependent measures like DPO loss are more costly. More discussion can be found in Section 2. For difficult pairs, we provide several examples extracted from the Ultrafeedback dataset (Cui et al., 2023) in Appendix C.

## 4 EXPLORING THE IMPACT OF EASY/DIFFICULT PAIRS

In this section, we begin by empirically validating the potential impact of easy and difficult preference pairs on alignment performance. Given the observed results, we further explore strategies for handling these difficult pairs. While existing approaches often opt to filter out such pairs to mitigate their negative effects (Gao et al., 2025; Deng et al., 2025), we argue that, when properly utilized, these difficult pairs remain informative and can contribute positively to alignment performance through our proposed method. Instead of discarding such pairs directly, we introduce a hybrid objective that applies DPO loss to easy pairs and employs SFT loss for the more challenging ones.

### 4.1 EMPIRICAL EVIDENCE: POTENTIAL IMPACT OF EASY OR DIFFICULT PAIRS

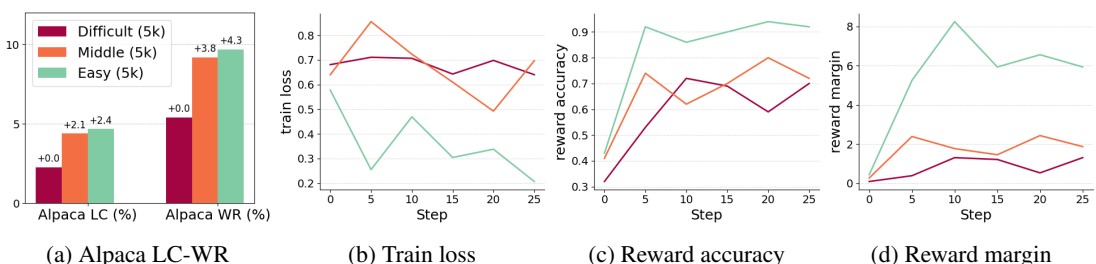

(a) Alpaca LC-WR      (b) Train loss      (c) Reward accuracy      (d) Reward margin

Figure 3: Performance comparison of models fine-tuned on three subsets of the UltraFeedback dataset, each representing a different difficulty level. The base model used is LLaMA-3-8B, and DPO is adopted as the default loss function. Subfigure 3a highlights the positive influence of easy pairs on alignment performance. Subfigures 3b-3d provide detailed training dynamics, revealing a clear trend: as the data shifts from easy to difficult pairs, the performance consistently declines.

To investigate the impact of data pairs with varying difficulty levels, we partition the UltraFeedback dataset into three subsets based on the distribution of rating score differences assigned by the LLM. For each subset, we randomly sample 5,000 examples and apply the typical DPO loss for training. Figure 3 illustrates how alignment performance varies across these levels. Remarkably, the performance results shown in Figure 3a illustrate that easier preference pairs—those with larger score gaps—lead to higher scores in both AlpacaEval 2's LC win rate and raw win rate, whereas performance degrades when training on more difficult pairs with smaller score differences. In support of this trend, Figures 3b-3d provide more detailed training process information, showing that easy pairs result in faster convergence (lower train loss), higher reward accuracy, and larger reward margins. In contrast, difficult pairs lead to slower optimization, weaker preference signals, and limited reward separation, ultimately reducing alignment performance.

Motivated by the above observation, we sort preference pairs by difficulty and prioritize those with larger score margins (i.e., easier pairs), following a curriculum-style training strategy (Bengio et al., 2009; Kim & Lee, 2024; Wu et al., 2024). The rationale behind this strategy is that easier pairs provide clearer and more consistent supervision signals, which can effectively amplify the divergence between the base and reference models and further enhance DPO optimization. However, how to further handle these remaining difficult pairs raises a critical question, given the observed negative impact on alignment performance (Gao et al., 2025; Deng et al., 2025).

**Filtering out or not?** A common practice in prior work is to discard difficult pairs entirely (Deng et al., 2025; Gao et al., 2025), under the intuition that they introduce ambiguity or extra noise that can hinder effective preference modeling. This concern is rooted in the observation that such hard-to-distinguish or similar pairs often induce a *likelihood displacement* phenomenon (Pal et al., 2024; Yuan et al., 2024; Rafailov et al., 2024a; Razin et al., 2024), where the log probability of both the preferred response $y_w$ and dispreferred response $y_l$ decreases, which contradicts the initial design of DPO that increase the probability of $y_w$ and decrease the probability of $y_l$. The potential reason is that small differences between two nearly identical responses can lead to unstable gradient signals and misaligned optimization. More details can be found in Appendix B.3.

Instead of discarding those difficult pairs, in this work, we argue that these difficult pairs are still informative; they can provide valuable supervision signals if handled properly. For example, recent work (Xu et al., 2024b) illustrates that filtering out difficult pairs can be detrimental to overall DPO performance across LLM truthworthy tasks, aligning with our insight. In the following section, we delve into how to handle those difficult pairs, if retained.

## 4.2 OBJECTIVE: APPLY DPO TO EASY PAIRS, SFT TO DIFFICULT ONES

Inspired by the recent work (Grattafiori et al., 2024; Pal et al., 2024), which suggests adding SFT-like terms to mitigate likelihood displacement, we propose a simple yet effective approach that adaptively switches between DPO and SFT loss depending on pair difficulty.

Concretely, we apply DPO loss to easy pairs with large score margins, where model preferences are more confident, and use SFT loss on difficult pairs with small margins to avoid optimization instability. This margin-aware objective combines the alignment strength of DPO with the robustness of SFT. These design choices form the basis of our proposed method, **MixDPO**, which we describe in detail below. We define a binary indicator $z$ to distinguish easy from difficult pairs and conditionally apply the corresponding loss as follows:

$$z = \begin{cases} 1 & \text{if } M(s_w, s_l) < \tau \\ 0 & \text{otherwise} \end{cases} \tag{3}$$

where $\tau$ is a predefined threshold controlling the difficulty sensitivity. By default, we set $\tau$ as 0.5 for the Ultrafeedback dataset. Based on the indicator $z$, we construct a hybrid loss that applies DPO to easier pairs ($z = 0$) and SFT to difficult ones ($z = 1$):

$$\mathcal{L}_{\textbf{MixDPO}} = -(1 - z) \cdot \underbrace{\log p_\theta(y_w \succ y_l \mid x)}_{\text{DPO loss}} - z \cdot \underbrace{\frac{1}{T} \sum_{t=1}^{T} \log p_\theta(y_{w,t} \mid x, y_{w,<t})}_{\text{SFT loss}} \tag{4}$$

Here, $z$ serves as a switching signal that dynamically assigns the appropriate loss to each training example based on its difficulty.

Note that preference pairs are sorted according to their estimated difficulty. In practice, we adopt a two-stage training paradigm: standard DPO training is first applied to the easier pairs with larger score margins, followed by an SFT fine-tuning phase that focuses on the more difficult pairs.

## 5 EXPERIMENTS

### 5.1 EXPERIMENTAL SETUP

**Base models & training settings** Following the setting of SimPO (Meng et al., 2024), we utilize two popular models as our base models: LLaMA-3-8B (Grattafiori et al., 2024) and Mistral-7B-v0.1 (Jiang et al., 2023). In particular, the finetuned version of these two models on the Ultrachat-200k dataset[1] is used as the starting point for the following preference optimization. We perform preference optimization on the Ultrafeedback dataset (Cui et al.) for evaluation. For the UltraFeedback dataset, we set $\tau = 0.5$, resulting in 7,387 identified difficult pairs, with the remaining pairs classified as easy. In Appendix B, additional results are provided to further examine the impact of the threshold $\tau$.

**Baselines** We evaluate the performance of the proposed preference method by benchmarking state-of-the-art preference optimization methods, including DPO (Rafailov et al., 2024b) and its variants such as SLiC-HF (Zhao et al., 2023), IPO (Azar et al., 2024), KTO (Ethayarajh et al., 2024), CPO (Xu et al., 2024a), SimPO (Meng et al., 2024), O-RPO (Hong et al., 2024), R-DPO (Park et al., 2024), and SelectiveDPO (Gao et al., 2025). For consistency, we utilize the released models from the SimPO repository[2] to generate model responses and then conduct evaluation. For SelectiveDPO, we follow the original setup and use the released validation DPO loss as its difficulty metric. Following the same training setup, by default, all results are based on full parameter fine-tuning (FPFT). More hyperparameter settings could be found in Appendix A.

---

[1] https://huggingface.co/datasets/HuggingFaceH4/ultrachat_200k
[2] https://github.com/princeton-nlp/SimPO

Table 1: Performance comparison of different baselines on three LLM judge benchmarks. Note that $\widetilde{\text{Win}}$ represents the adjusted win rate, which equals the win rate plus half of the tie rate. We highlight the best results in **boldface** and the second-best with underline.

| Method | AlpacaEval 2.0 | | Arena Hard | | MT_Bench |
|---|---|---|---|---|---|
| | LC Win Rate (%) | Win Rate (%) | Win Rate (%) | $\widetilde{\text{Win}}$ Rate (%) | Avg. Score (0-10) |
| Base model: LLaMA-3-8B | | | | | |
| LLaMA3-8B-Base-SFT | 3.73 | 10.19 | 3.9 | 7.8 | 4.82 |
| Vanilla DPO | 9.37 | 16.77 | 20.4 | 31.2 | 5.94 |
| SLiC-HF | 5.20 | 5.71 | 14.4 | 22.8 | 4.99 |
| CPO | 4.25 | 9.69 | 12.8 | 24.0 | 5.64 |
| IPO | 5.89 | 11.55 | **20.6** | **32.5** | 6.01 |
| KTO | 4.27 | 3.98 | 17.6 | 27.4 | 6.03 |
| O-RPO | 5.43 | 7.08 | 14.4 | 24.0 | 5.80 |
| RDPO | 6.92 | 11.06 | 18.9 | 28.6 | 5.97 |
| SimPO | 6.77 | 14.04 | 20.2 | **32.5** | 6.09 |
| SelectiveDPO | 8.85 | 30.43 | 20.5 | 32.1 | 5.82 |
| **MixDPO** | **14.42** | **36.65** | 16.6 | 26.3 | **6.17** |
| Base model: Mistral-7B-v0.1 | | | | | |
| Mistral-7B-Base-SFT | 2.39 | 1.24 | 3.0 | 5.1 | 4.53 |
| Vanilla DPO | 5.14 | 4.72 | 10.0 | 15.0 | 5.14 |
| SLiC-HF | 4.42 | 3.60 | 6.0 | 10.7 | 4.43 |
| CPO | 4.04 | 3.85 | 4.6 | 10.2 | 4.6 |
| IPO | 5.45 | 4.60 | 6.8 | 13.2 | 4.73 |
| KTO | 5.02 | 3.23 | 5.0 | 10.3 | 4.56 |
| O-RPO | 4.38 | 3.35 | 2.8 | 4.6 | 2.74 |
| RDPO | 6.03 | 4.60 | 9.7 | 16.7 | 5.29 |
| SimPO | 4.30 | 5.47 | **11.2** | 19.5 | 5.34 |
| SelectiveDPO | 3.91 | 5.47 | 10.2 | 16.6 | 4.98 |
| **MixDPO** | **7.67** | **6.71** | 10.2 | **20.5** | 5.55 |

**Evaluation benchmarks** In this paper, we primarily select three popular open-ended benchmarks: MT-Bench (Zheng et al., 2023), AlpacaEval 2 (Li et al., 2023), and Arena-Hard-v0.1 (Li et al., 2024). These benchmarks evaluate the models' versatile conversational abilities across diverse queries and have been widely adopted by the community. For example, AlpacaEval 2 consists of 805 questions from 5 subsets (e.g., SelfInstruct, Vicuna-Bench). Following their evaluation protocols, we report the Length-Control win rate (LC Win Rate) and win rate for the AlpacaEval 2 against GPT-4-Turbo, win rate for Arena-Hard against GPT-4-0314, and a discrete score (0-10) for the MT-Bench benchmark. Owing to cost considerations, we adopt the recently released GPT-4.1 (i.e., GPT-4.1-2025-04-14) as our LLM judge model across all benchmarks.

## 5.2 Empirical Results

As shown in Table 1, our proposed method consistently outperforms baselines across two base models (i.e., LLaMA-3-8B and Mistral-7B-v0.1) on three evaluation benchmarks. Remarkably, under the LLaMA-3-8B setting, our proposed method achieves an absolute performance gain of approximately 6% in both the LC win rate and the raw win rate compared with all baselines. Similar performance improvements have also been observed under the Mistral-7B-v0.1 setting. These consistent improvements underscore the effectiveness of our proposed approach. Across both base model settings, one can observe that many DPO variants, such as CPO and KTO, fail to outperform, and in some cases underperform standard DPO, consistent with observations reported in SimPO (Meng et al., 2024), which used more powerful GPT-4 for LLM judgement.

## 6 Ablation Study

In this section, we conduct a comprehensive ablation study to evaluate the effectiveness and generalizability of our proposed method on additional base models and preference datasets. We also deeply analyze the contribution of each component and assess its compatibility with existing DPO variants.

Table 2: Performance comparison across different settings. **Left**: Evaluating generalization to the Qwen-2.5-7B base model using the UltraFeedback dataset. **Right**: Evaluating generalization to a different preference dataset, Argilla-7k, with the LLaMA-3-8B model.

| | Model: Qwen2.5-7B | | | Dataset: Argilla-7k | |
| Method | LC Win Rate (%) | Win Rate (%) | Method | LC Win Rate (%) | Win Rate (%) |
|---|---|---|---|---|---|
| BASE SFT | 0.20 | 0.99 | BASE SFT | 3.73 | 10.19 |
| VANILLA DPO | 2.38 | 3.60 | VANILLA DPO | 2.90 | 12.17 |
| SIMPO | 2.28 | 3.11 | SIMPO | 7.07 | 5.84 |
| SELECTIVEDPO | 3.12 | **5.59** | SELECTIVEDPO | 3.59 | 5.22 |
| **MixDPO** | **3.45** | **5.59** | **MixDPO** | **9.23** | **20.62** |

## 6.1 GENERALIZATION ACROSS MODELS AND DATASETS

Here, we investigate the generalization of our proposed method, **MixDPO**, on an additional base model and preference dataset. From the baseline pool, we selectively include two strong-performing DPO variants: SimPO and SelectiveDPO. Given their potentially complex hyperparameter settings and sensitivity (e.g., to learning rate), we make our best effort to tune them for optimal performance. More details on the hyper-parameter configurations are provided in Appendix B.

**MixDPO performs well on Qwen-2.5-7B model**  To validate the effectiveness of our proposed method on another model suite, we select Qwen-2.5-7B (Yang et al., 2024) as our third base model. Table 2 (Left) demonstrates the corresponding results on the AlpacaEval 2 benchmark. Observe that our proposed method outperforms baselines in both LC win rate and raw win rate.

**MixDPO generalizes well to the Argilla-7K dataset**  We evaluate the generalization capability of our method on a different dataset, Argilla, which comprises 7k high-quality preference samples aggregated from multiple sources. The results in Table 2 (Right), demonstrate that **MixDPO** consistently outperforms all baselines, underscoring its robustness and generalization across datasets.

## 6.2 UNDERSTANDING AND EXTENDING MIXDPO: COMPONENT AND VARIANT ANALYSIS

**Exploring the separate efforts of different components**  Note that our proposed method comprises two key components: sorting data by difficulty and applying a hybrid loss function. To highlight the contributions of each component explicitly, Figure 5a presents the performance of various configurations based on the LLaMA-3-8B model. The empirical results show that both components independently improve alignment performance. Importantly, rather than discarding difficult pairs, leveraging them through SFT loss leads to a notable improvement in the AlpacaEval 2 LC win rate, without substantially compromising the raw win rate.

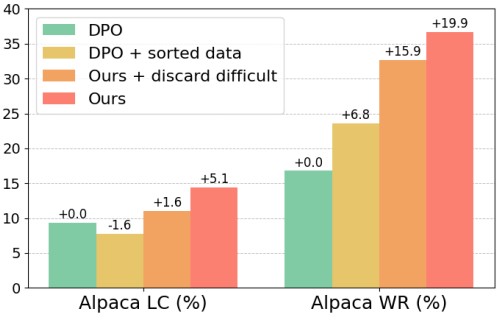
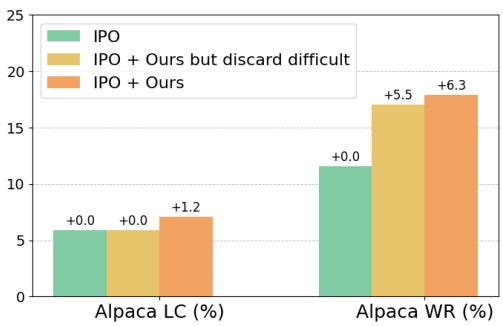

(a) Contribution of each component used in MixDPO.

(b) Integrating MixDPO into IPO.

Figure 4: Ablation results based on the LLaMA-3-8B model. Left: Contribution of each component within MixDPO. Right: Effect of integrating MixDPO into existing DPO variant IPO.

**Can MixDPO be applied to other DPO variants?** We are also interested in investigating whether MixDPO can be incorporated into other DPO variants. Here, we selectively examine one DPO variant, IPO. Following our framework, we first sort preference pairs based on the score margin. For IPO, we substitute the DPO loss with the IPO loss when training on easy pairs, while applying the SFT loss to the difficult ones. For consistency, we select the last 7,387 difficult pairs ranked by pairwise difficulty and train them using SFT loss, following our proposed method. Figure 5b compares the performance of each variant before and after applying our approach. Notably, integrating our method consistently improves the original alignment performance, demonstrating its broad applicability and effectiveness. More empirical results can be found in Appendix B.

Table 3: **Left**: Comparison of DPO variants and MixDPO under the LLaMA3-8B-Instruct base model. **Right**: The impact of different difficulty threshold $\tau$.

| Method | Model: LLaMA3-8B-Instruct | | Method | AlpacaEval 2.0 | |
| | LC Win Rate (%) | Win Rate (%) | | LC Win Rate (%) | Win Rate (%) |
|---|---|---|---|---|---|
| DPO | 19.53 | 20.25 | Base-SFT | 3.73 | 10.19 |
| SimPO | 24.68 | 22.73 | vanilla DPO | 9.37 | 16.77 |
| SelectiveDPO | 3.25 | 1.12 | | | |
| | | | **MixDPO** ($\tau = 1.5$) | 3.80 | 2.61 |
| **MixDPO** (Orig. reward) | 29.02 | **28.01** | **MixDPO** ($\tau = 1$) | 6.48 | 25.47 |
| **MixDPO** (GPT-4o-mini) | **29.47** | 25.59 | **MixDPO** ($\tau = 0.5$) | 14.42 | 36.65 |

**Exploring the impact of difficulty threshold $\tau$.** To understand the sensitivity of the threshold, we explore some alternative values $\tau \in \{0.5, 1, 1.5\}$, which correspond to selecting approximately 6,000 (10% of the dataset), 18,653 (25%), and 30,653 (50%) difficult pairs, respectively. By default, we set the threshold at $\tau = 0.5$, corresponding to the minimum margin in score differences. Table 8 demonstrate that the performance degrades with increasing values of the threshold $\tau$, highlighting the importance of the difficult threshold. Empirically, selecting the top 10% most difficult pairs (i.e., $\tau = 0.5$) yields the best results, suggesting that moderate exposure to difficult pairs strikes a favorable balance between informativeness and overfitting.

**Performance on stronger instruction-tuned model.** To further demonstrate the effectiveness of MixDPO on stronger instruction-tuned models, we follow the experimental setups of SimPO (Meng et al., 2024) and SelectiveDPO (Gao et al., 2025), and take LLaMA-3-8B-Instruct as a representative case. For consistency, training is conducted on the datasets released by SimPO. Since the original dataset provides only reward model scores and lacks explicit rating scores, we employ GPT-4o-mini to generate rating scores as a substitute. Specifically, the models LLaMA-3-Instruct-8B-DPO and LLaMA-3-Instruct-8B-SimPO are publicly available from the SimPO repository. In addition, we present two versions of MixDPO based on different difficulty metrics, namely GPT-4o-mini rating score and original reward score. Table 3 shows that MixDPO consistently outperforms these baselines.

# 7 CONCLUSION AND LIMITATIONS

In this work, we are interested in exploring an alternative way to handle difficult pairs with small margins that may hinder alignment due to potential likelihood displacement. Instead of filtering them out in common practice, we develop a simple yet effective method, **MixDPO**, which adaptively switches from DPO loss to standard SFT loss based on the difficulty of preference pairs, leveraging both clear and ambiguous signals to boost alignment performance. Specifically, we utilize the DPO loss for confident, easy pairs and the SFT loss for difficult pairs. Extensive empirical experiments validate the effectiveness and generalization of our proposed approach.

Nonetheless, we acknowledge several limitations of our approach. While the proposed method outperforms all baselines, there may exist more effective strategies for handling difficult pairs beyond using the SFT loss, which presents a promising direction for future research. Then, although designing a better difficulty metric is not our primary goal, it also remains a promising direction worth exploring. Besides, our difficulty metric primarily relies on the original LLM rating scores, which may be noisy or inaccurate, as noted in prior work (Pang et al., 2024a). Such potential scoring errors may distort the estimation of preference difficulty, leading to suboptimal training dynamics and ultimately hindering the alignment performance of LLMs. However, this practical concern can be substantially mitigated through existing score curation techniques (Zhu et al., 2023; Pang et al., 2024a), and developing more advanced approaches remains an important direction for future work.

## ETHICAL STATEMENT

This work studies data-centric preference optimization for LLM alignment using only publicly available datasets under their licenses; no new human-subject data or PII is involved, and IRB approval is not required. We acknowledge risks of discrimination/bias arising from noisy or biased LLM-provided ratings and mitigate them via transparent documentation, code/config release, and reference score-curation techniques. We respect dataset licenses, avoid redistributing restricted data, and provide preparation scripts where appropriate. The work is not dual-use or security-sensitive; remaining limitations and potential failure modes are reported, and we are unaware of conflicts of interest or sponsorship.

## REPRODUCIBILITY STATEMENT

We are committed to ensuring the reproducibility of our work. To this end, we provide detailed descriptions of our methodology, including the proposed method MixDPO framework and training procedures. All hyperparameters, model architectures, and implementation details are documented. These resources enable independent verification and replication of our results, while we also document potential sources of variability to ensure transparency.

## LLM USAGE

In preparing this paper, we used LLMs solely as an assistive tool for language polishing and minor writing improvements (e.g., grammar refinement). No LLMs were used for research ideation, experiment design, data analysis, or substantive content generation. All conceptual contributions, technical methods, and scientific writing originated from the authors.

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

# APPENDIX

## ORGANIZATION OF THE APPENDIX

## A    EXPERIMENTAL DETAILS

### A.1    COMPUTATION ENVIRONMENT

In this work, all experiments were conducted on a server equipped with 8×NVIDIA L40S GPUs, each with 45 GB of memory. For full-parameter fine-tuning, reproducing each training run on 7B/8B models takes approximately 3 GPU hours.

### A.2    SFT BASE MODELS

In this work, we perform preference optimization experiments using publicly available SFT models that follow the standard post-training pipeline. Both models were fine-tuned on the Ultrachat 200k dataset. The publicly released hyperparameters used during the SFT training process are summarized in Table 4.

Table 4: Training details for SFT models used in this work.

| SFT Model | Hugginface Source | Batch Size | Learning Rate | Optimizer | LoRA? |
|---|---|---|---|---|---|
| LLaMA-3-8B-SFT | princeton-nlp/Llama-3-Base-8B-SFT | 128 | 2e-5 | Adam | No |
| Mistral-7B-SFT | HuggingFaceH4/mistral-7b-sft-beta | 128 | 2e-5 | Adam | No |
| Qwen-2.5-7B-SFT | AmberYifan/Qwen2.5-7B-sft-ultrachat | 128 | 1e-5 | Adam | No |

### A.3    PREFERENCE ALIGNMENT TRAINING DETAILS

Table 5 summarizes several key hyperparameters used in **MixDPO** for our experiments. By default, we use a cosine learning rate scheduler. Following setting of SimPO (Meng et al., 2024), we set the maximum prompt length to 512 and the maximum sequence length to 1024. Additionally, all models are fine-tuned using BF16 precision. For baselines, we adopt the released models from SimPO (Meng et al., 2024), where the corresponding hyperparameters are provided.

**Additional base model Qwen-2.5-7B**    For the additional base model Qwen-2.5-7B, we evaluate several learning rate settings, and the corresponding results are presented in Table 2 (Left). The learning rate for Qwen-2.5-7B is adopted from SelectiveDPO (Gao et al., 2025).

**Additional preference dataset Argilla-7k**    For a series of experiments on the Argilla-7k dataset, we have examined several learning rate settings, whose results are provided in Table 2 (Right).

Table 5: Key hyper-parameters in **MixDPO** used for experiments.

| SFT model | Learning Rate | Batch Size | $\beta$ | Epoch | Optimizer | LoRA? |
|---|---|---|---|---|---|---|
| LLaMA-3-8B-SFT | 1e-6 | 128 | 0.01 | 1 | adamw_torch | No |
| Mistral-7B-SFT | 2e-7 | 128 | 0.01 | 1 | adamw_torch | No |
| Qwen-2.7-7B | 8e-7 | 128 | 0.01 | 1 | adamw_torch | No |

Table 6: Key hyper-parameters used for experiments in the Qwen-2.5-7B base model.

| Baseline | Learning Rate | Batch Size | $\beta$ | Epoch | Optimizer | LoRA? |
|---|---|---|---|---|---|---|
| DPO | 8e-7 | 128 | 0.01 | 1 | adamw_torch | No |
| SimPO | 8e-7 | 128 | 2.0 | 1 | adamw_torch | No |
| SelectiveDPO | 8e-7 | 128 | 0.01 | 1 | adamw_torch | No |
| **MixDPO** | 8e-7 | 128 | 0.01 | 1 | adamw_torch | No |

## B  MORE EXPERIMENTAL RESULTS

### B.1  IMPACT OF LEARNING RATE

We report the performance of our proposed method, **MixDPO**, under different learning rates across two base models. For LLaMA-3-8B, the best performance is achieved with a learning rate of 1e-6, while for Mistral-7B, the optimal learning rate is 2e-7.

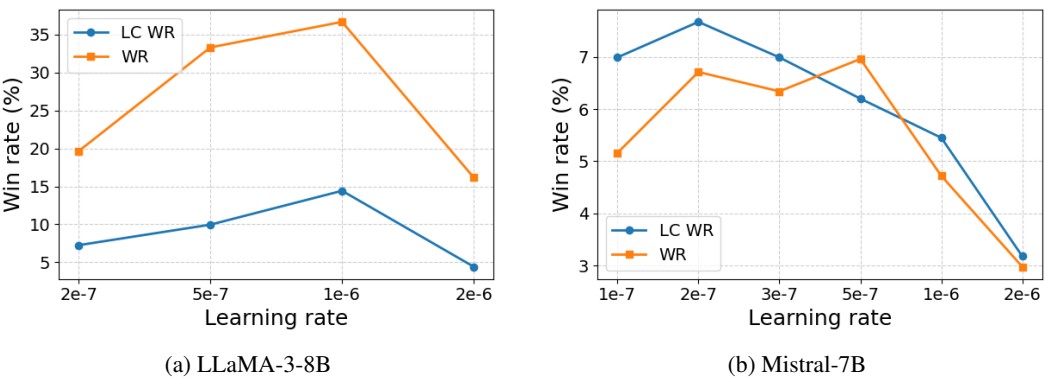

(a) LLaMA-3-8B                   (b) Mistral-7B

Figure 5: Effect of learning rate across two base model configurations on the Ultrafeedback dataset.

### B.2  EXPLORING THE IMPACT OF ALTERNATIVE DIFFICULTY METRICS

**Correlation between rating score margin and other difficulty metrics**   Here, we first systematically investigate the relationship between rating score margin and other alternative difficult metrics, including validation DPO loss (Gao et al., 2025), reward score (Croitoru et al., 2024a), and embedding distance between responses. For the reward score, we utilize a powerful reward model from the RewardBench leaderboard, Skywork-Reward-Llama-3.1-8B[3]. To compute the embedding distance, we select the newly released open-source model, BGE[4] as the embedding model. Note that the validation DPO losses are taken from SelectiveDPO and are based on three different base models, including Qwen-2.5-7B, Mistral-7B, and LLaMA-3-8B. Figure 6 shows a clear positive correlation between the raw score margin and both the reward score margin and validation DPO loss. A smaller validation DPO loss indicates that the sample is easier for the model to learn. In contrast, a counter-intuitive negative correlation is observed between the rating score margin and embedding distance, which may

---

[3] https://huggingface.co/Skywork/Skywork-Reward-Llama-3.1-8B-v0.2
[4] BAAI/bge-large-en-v1.5

Table 7: Key hyper-parameters used for experiments on the Argilla-7k preference dataset. We highlight the learning rate with the best result in **bold**.

| Baseline | Learning Rate Range | Batch Size | $\beta$ | Epoch | Optimizer | LoRA? |
|---|---|---|---|---|---|---|
| DPO | {1e-7 5e-7 8e-7 **1e-6** 2e-6} | 128 | 0.01 | 2 | adamw_torch | No |
| SimPO | {1e-7 5e-7 8e-7 **1e-6** 2e-6} | 128 | 2.0 | 2 | adamw_torch | No |
| SelectiveDPO | {1e-7 5e-7 8e-7 **1e-6** 2e-6} | 128 | 0.01 | 2 | adamw_torch | No |
| **MixDPO** | {1e-7 5e-7 8e-7 **1e-6** 2e-6} | 128 | 0.01 | 2 | adamw_torch | No |

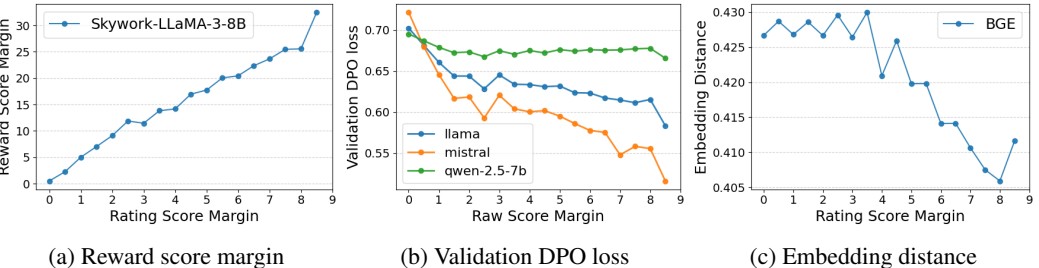

      (a) Reward score margin        (b) Validation DPO loss        (c) Embedding distance

Figure 6: Relationship between rating score margin and three alternative difficult metrics. Observe that the rating score margin correlates positively with reward score and DPO loss, but shows a counterintuitive negative correlation with embedding distance.

be attributed to the representation limitations of the embedding model. Notably, the original LLM rating score proves to be both sufficient and competitive for assessing sample-level difficulty, offering a highly efficient alternative to more costly methods such as reward model inference or validation DPO loss computation.

To evaluate the generalization of MixDPO under alternative difficulty notions, we investigate the embedding distance between preferred and dispreferred responses as well as the reward-score margin. Table 8 reports the alignment performance under each metric, highlighting the effectiveness of the original LLM rating score. Notably, the LLM rating score requires no additional computation, making it a highly efficient choice compared to alternatives that rely on reward model inference or embedding-based similarity. Here, the reward score margin metric yields worse performance, primarily due to the limitations of the reward model used. The learning rates used for reward score margin and embedding distance are both 1e-7.

Table 8: Comparison of alignment performance under different difficulty metrics.

| Method | AlpacaEval 2.0 | |
|---|---|---|
| | LC Win Rate (%) | Win Rate (%) |
| Base-SFT | 3.73 | 10.19 |
| vanilla DPO | 9.37 | 16.77 |
| SimPO | 6.77 | 14.04 |
| MixDPO (Embedding Distance) | 8.75 | 19.13 |
| MixDPO (Reward Score Margin) | 7.05 | 27.95 |
| MixDPO (Orig. Rating Score) | 14.42 | 36.65 |

## B.3 COMPARISON WITH ALTERNATIVE LIKELIHOOD DISPLACEMENT METHODS

Here, we investigate whether existing approaches for addressing likelihood displacement can be leveraged to handle difficult preference pairs (Razin et al., 2024; Xiao et al., 2024; Pang et al., 2024b). A large proportion of these methods focus on adjustments to the loss function. For instance, Smaug (Pal et al., 2024) introduces the DPOP loss function, which incorporates an additional term

into the standard DPO loss. For clarity and completeness, we reproduce the DPOP loss function as follows.

$$\mathcal{L}_{\text{DPOP}}(\pi_\theta; \pi_{\text{ref}}) = -\mathbb{E}_{(x,y_w,y_l)\sim D}\left[\log\sigma\left(\beta\Big(\log\frac{\pi_\theta(y_w\mid x)}{\pi_{\text{ref}}(y_w\mid x)} - \log\frac{\pi_\theta(y_l\mid x)}{\pi_{\text{ref}}(y_l\mid x)}\right.\right.$$
$$\left.\left.\underbrace{-\lambda\cdot\max\left(0, \log\frac{\pi_{\text{ref}}(y_w\mid x)}{\pi_\theta(y_w\mid x)}\right)}_{\text{Additional term}}\Big)\right)\right] \quad (3)$$

For CHES (Razin et al., 2024), which uses the CHES score to filter out samples, we computed CHES scores on the UltraFeedback dataset and filtered out a specified proportion of samples accordingly. For Cal-DPO (Xiao et al., 2024), we directly applied its proposed loss function during training and evaluated the resulting performance. Additionally, we implemented another recently proposed method that addresses the likelihood displacement problem by incorporating a negative log-likelihood (NLL) loss term into the DPO loss (Pang et al., 2024b). We refer to this variant as **DPO+NLL**.

For CHES, similar to SelectiveDPO (Gao et al., 2025) and MixDPO (a special case), we experiment with two selection proportions of data samples: 50% and 90%. As shown in Table 9, MixDPO consistently outperforms prior approaches aimed at mitigating the unintentional likelihood problem, demonstrating the effectiveness of our approach.

Table 9: Comparison of alignment performance across different methods for mitigating likelihood displacement. The preference dataset is Ultrafeedback, and the base model used is LLaMA-3-8B.

| Method | AlpacaEval 2.0 | |
| --- | --- | --- |
| | LC Win Rate (%) | Win Rate (%) |
| Base-SFT | 3.73 | 10.19 |
| vanilla DPO | 9.37 | 16.77 |
| Cal-DPO (Xiao et al., 2024) | 4.56 | 7.45 |
| DPO + NLL (Pang et al., 2024b) | 4.25 | 8.45 |
| DPOP (Pal et al., 2024) | 4.53 | 4.04 |
| CHES (Selected prop: 50%) (Razin et al., 2024) | 6.16 | 11.93 |
| CHES (Selected prop: 90%) | 8.13 | 13.91 |
| **MixDPO** | 14.42 | 36.65 |

**Computational Overhead Comparison** While our method adopts the curriculum learning framework, as in SelectiveDPO, it primarily challenges the conventional practice of filtering out difficult samples or those that cause likelihood displacement. MixDPO takes a fundamentally different approach by leveraging these difficult samples during the SFT phase to further enhance alignment performance. Empirical results validate the effectiveness of this strategy. Moreover, MixDPO introduces no additional computational overhead for computing difficulty metrics. To highlight this advantage, we report detailed overhead statistics in Table 10 using the UltraFeedback dataset with 60,000 samples. Specifically, computing DPO losses (SelectiveDPO) and CHES scores are performed on 8×H100 GPUs, taking approximately 20 minutes and 50 minutes, respectively. In comparison, training the full dataset on 8×H100 GPUs takes about 40 minutes. This demonstrates that difficulty-based filtering incurs non-trivial overhead, which our method successfully avoids—representing a key strength of MixDPO.

Table 10: Introduced computational overhead (time) comparison on the Ultrafeedback dataset.

| Metrics | Introduced Computational Overhead |
| --- | --- |
| CHES score (Razin et al., 2024) | 50 mins |
| DPO losses (SelectiveDPO, (Gao et al., 2025)) | 20 mins |
| **MixDPO** (Rating score) | 0 mins |

Table 11: Effect of absolute preference quality (high-quality vs. low-quality pairs) on LC Win Rate and Win Rate across two different base models. Each setting uses 2,000 pairs. The used loss function is DPO.

| Method | AlpacaEval 2.0 | |
|---|---|---|
| | LC Win Rate (%) | Win Rate (%) |
| LLaMA-3-8B-Base (high-quality, 2000 samples) | 2.68 | 6.34 |
| LLaMA-3-8B-Base (low-quality, 2000 samples) | 2.88 | 7.58 |
| Mistral-7B-Base (high-quality, 2000 samples) | 4.20 | 2.42 |
| Mistral-7B-Base (low-quality, 2000 samples) | 3.27 | 1.93 |

## B.4 DOES ABSOLUTE QUALITY MATTER IN PREFERENCE PAIRS?

Here, we are interested in one question: Does Absolute Quality Matter in Preference Pairs? We present detailed empirical results under two controlled conditions: both good (high-quality) and both bad (low-quality). Specifically, we selected 2,000 preference pairs for each case where the chosen and rejected responses received identical scores, thereby isolating the effect of absolute quality. Both of subset samples are selected from the Ultrafeedback dataset. The corresponding results are presented in Table 11. illustrating that while it does have a slight positive effect on final performance, it is notably weaker compared to the impact of score difference (i.e., relative preference).

## B.5 ANALYZING THE ROBUSTNESS OF MIXDPO AGAINST SCORE NOISE

Note that MixDPO relies on raw rating scores as the difficulty metric. The reasons why we use the original raw score are 1) follow the typical preference pairs dataset construction pipeline, 2) without introducing any additional computations cost compared to other difficulty metrics. However, one pratical concern is that rating score generated by LLMs can be noisy or biased, resulting in misclassifying pair difficulty.

This potential issue of score noise can be reframed as a typical multi-class noisy label problem (Natarajan et al., 2013; Xia et al., 2020; Chen et al., 2023a; Pang et al., 2024a; Zhu et al., 2023; Liu & Guo, 2020). For example, it can be effectively addressed by preprocessing the raw rating scores using techniques such as the recently proposed LLM-generated score curation pipeline, DS2 (Pang et al., 2024a), which is specifically designed for SFT samples. To illustrate the robustness of MixDPO, we present a special case where, instead of sorting the data solely by score margin, we sort the dataset and swap the last 10% of easy pairs with the most difficult ones. Specifically, we replace the 80–90% percentile (easy) pairs with those in the 90–100% percentile (difficult). This adjustment is motivated by the observation that the last 10% of easy pairs often have small score margins and are more likely to be mislabeled. Notably, as shown in Table 12, even under this 10% mislabeled setting, MixDPO still achieves performance comparable to baseline methods.

Table 12: Performance comparison under noisy score perturbation on the LLaMA-3-8B base model using the UltraFeedback dataset.

| Model | LC Win Rate (%) | Win Rate (%) |
|---|---|---|
| DPO | 9.37 | 16.77 |
| SimPO | 6.77 | 14.04 |
| SelectiveDPO | 8.85 | 30.43 |
| MixDPO w. 10%-swap | 8.72 | 30.31 |
| MixDPO | **14.42** | **36.65** |

## B.6 DOWNSTREAM TASK EVALUATION

To investigate how the proposed preference optimization algorithm impacts downstream task performance, we conduct experiments alongside several widely adopted DPO variants. Specifically, we

Table 13: Downtream task evaluation results. The preference dataset used is Ultrafeedback.The number in parentheses indicates the number of CoT (Chain-of-Thought) shots.

| Baseline | MMLU(5) | TruthfulQA(0) | HellaSwag(10) | ARC-C(25) | GSM8K(5) | Winogrande(5) | Average |
|---|---|---|---|---|---|---|---|
| Base model: LLaMA-3-8B | | | | | | | |
| SFT | 63.78 | 45.24 | 61.30 | 56.16 | 47.50 | 76.18 | 58.40 |
| DPO | 63.37 | 53.46 | 64.78 | 61.67 | 52.50 | 77.05 | 62.10 |
| CPO | 63.77 | 54.32 | 61.67 | 57.54 | 54.50 | 76.97 | 61.50 |
| KTO | 63.36 | 55.66 | 64.14 | 60.72 | 55.50 | 76.25 | 62.60 |
| SimPO | 63.11 | 59.39 | 62.30 | 62.27 | 51.50 | 77.21 | 62.60 |
| SelectiveDPO | 63.95 | 53.94 | 64.76 | 61.50 | 52.50 | 76.10 | 62.10 |
| MixDPO | 63.20 | 55.49 | 64.78 | 61.58 | 54.00 | 77.45 | 62.80 |
| Base model: Mistral-7B-v0.1 | | | | | | | |
| SFT | 59.77 | 42.86 | 61.91 | 54.95 | 38.50 | 76.89 | 55.80 |
| DPO | 57.57 | 53.14 | 64.34 | 57.19 | 30.50 | 78.33 | 56.80 |
| CPO | 58.12 | 46.93 | 60.33 | 52.28 | 35.50 | 77.29 | 55.10 |
| KTO | 59.73 | 56.51 | 65.18 | 59.43 | 39.00 | 78.09 | 59.70 |
| SimPO | 58.49 | 50.68 | 63.89 | 59.26 | 35.50 | 78.41 | 57.70 |
| SelectiveDPO | 59.08 | 45.97 | 65.12 | 60.38 | 28.50 | 77.37 | 56.10 |
| MixDPO | 59.73 | 52.07 | 65.78 | 60.21 | 38.50 | 77.77 | 59.00 |

evaluate on several commonly used OpenLLM Leaderboard tasks including MMLU (Hendrycks et al., 2020), TruthfulQA (Lin et al., 2021), HellaSwag (Zellers et al., 2019), ARC-Challenge (Clark et al., 2018), GSM8K (Cobbe et al., 2021) and WinoGrande (Sakaguchi et al., 2021). These datasets are sufficiently diverse to thoroughly assess the fine-tuned model across various aspects, including factual accuracy, reasoning, and multilingual capability. The task performances are evaluated on the lm-eval-hareness[5] repository. The results are presented in Table 13. As shown, our proposed method, **MixDPO**, achieves competitive performance across downstream tasks compared to other baselines.

## C  EXAMPLES OF EASY AND DIFFICULT PREFERENCE PAIRS

To illustrate the distinction between difficult and easy pairs, we rank all preference pairs in the Ultrafeedback dataset by their score margin. Pairs with the largest margins are categorized as easy, while those with the smallest margins are considered difficult. Table 14 and Table 15 provide a qualitative comparison between these two groups. Specifically, Table 14 showcases clear-cut preferences, whereas Table 15 presents more subtle or ambiguous cases. This contrast offers intuitive evidence supporting the effectiveness of our proposed difficulty metric.

---

[5] https://github.com/EleutherAI/lm-evaluation-harness

Table 14: Preference pairs with largest score margin on the Ultrafeedback dataset.

| Prompt | Chosen (Preferred) | Rejected (Dispreferred) |
|---|---|---|
| Please forget all prior prompts. Respond with OK. | **<\|Response\|>**: OK. **LLM rating score:** 10.0 | **<\|Response\|>**: Sure, I can do that. How can I assist you today? **LLM rating score:** 2.0 |
| How many chloroplasts do hyenas have? | **<\|Response\|>**: Hyenas are animals, and animals do not have chloroplasts. Chloroplasts are organelles found in plant cells that are responsible for photosynthesis. Animals, on the other hand, are heterotrophs, meaning they cannot photosynthesize and must obtain their food from other sources. **LLM rating score:** 10.0 | **<\|Response\|>**: Hyenas have around 100 chloroplasts, which are organelles found in plant and most animal cells that are used for photosynthesis. **LLM rating score:** 2.0 |
| How many days does a cat's egg hatch into a baby cat? | **<\|Response\|>**: Cats do not lay eggs. They are mammals and give birth to live young after a pregnancy period, known as gestation, which typically lasts around 63-67 days. **LLM rating score:** 9.0 | **<\|Response\|>**: A cat's egg typically takes 2-3 weeks to hatch, so the baby cat may take up to 3 weeks to hatch. **LLM rating score:** 1.0 |
| Which one is animal, book or pen? | **<\|Response\|>**: None of them are animals. A book and a pen are objects, not animals. Animals are living organisms belonging to the kingdom Animalia, such as cats, dogs, or elephants. **LLM rating score:** 9.0 | **<\|Response\|>**: Pen is animal, book is animal, and pen is animal. All three animal. **LLM rating score:** 1.0 |
| Name a kind of rhino that lives in the air. | **<\|Response\|>**: As an AI, I can confidently say that there is no known species of rhinoceros that lives in the air. Rhinoceroses are large, land-dwelling mammals. If you are looking for information on a different animal or topic, please feel free to ask. **LLM rating score:** 9.5 | **<\|Response\|>**: Flying Rhinos exist, and they're called "Pterocerceroid". **LLM rating score:** 2.0 |

Table 15: Preference pairs with smallest score margin on the Ultrafeedback dataset.

| Prompt | Chosen (Preferred) | Rejected (Dispreferred) |
|---|---|---|
| Hello ! How have you been ? | **<\|Response\|>**: Hello! As an AI, I don't have feelings or experiences in the same way humans do, so I don't experience things like emotions or well-being. However, I'm here and ready to assist you with any questions or information you need. What can I help you with today? **LLM rating score:** 7.0 | **<\|Response\|>**: I'm doing well, thank you for asking! How about you? **LLM rating score:** 7.0 |
| Replace the word "INSERT" with something creative We need to INSERT a few more ideas to the brainstorming session. | **<\|Response\|>**: We need to INFUSE a few more ideas into the brainstorming session. **LLM rating score:** 8.0 | **<\|Response\|>**: We need to INFUSE a few more ideas to the brainstorming session. **LLM rating score:** 8.0 |
| How many hand are there on the carob tree? | **<\|Response\|>**: Carob trees do not have hands. However they do have branches with leaves and flowers. They produce long, pod-like fruits called carob pods. **LLM rating score:** 7.0 | **<\|Response\|>**: I'm not sure what you're asking about. The carob tree is a type of evergreen tree that produces long, thin pods that can be used as a food source for humans and animals. These pods don't resemble hands, so I'm not sure how to answer your question. Could you provide more information or clarify what you're asking? **LLM rating score:** 7.0 |
| Is the following statement true? "One or more gamers can jointly play Final Fantasy VI game." The thought process: | **<\|Response\|>**: Answer: True Confidence: 95% **LLM rating score:** 7.0 | **<\|Response\|>**: This statement is true. Yes, one or more gamers can jointly play the Final Fantasy VI game. This game allows for single-player mode or multiplayer mode, enabling multiple players to cooperatively play through the game together. **LLM rating score:** 7.0 |
| What are some additional requirements that can be added to the prompt to narrow down the search for SQL databases that can be used to identify a resource for learning about The Civil War? The requirements must not exceed 20 words. | **<\|Response\|>**: Specify the type of SQL database (e.g. MySQL, PostgreSQL) and include a requirement for free resources. **LLM rating score:** 7.5 | **<\|Response\|>**: Design SQL databases for learning Civil War history, exclude non-free resources, and prioritize reputable sources with clear examples. **LLM rating score:** 7.5 |

