# OpenReview forum: "Supervised Fine-Tuning on Ambiguous Preference Pairs Boosts LLM Alignment"
_ICLR.cc/2026/Conference — ICLR 2026 Conference Withdrawn Submission_

### Official Review · Reviewer_KCfS · 2025-10-25

**Soundness:** 2
**Presentation:** 2
**Contribution:** 2
**Rating:** 2
**Confidence:** 4

**Summary:**

This paper studies how to utilize ambiguous preference pairs in LLM alignment.
The authors argue that the ambiguous and "difficult" preference pairs, where the predicted scores of the two responses are close, still contain useful information for alignment.
Unlike prior works that discard such pairs, the authors propose to apply a curriculum learning strategy that sorts the training data by difficulty and gradually introduces more difficult pairs during DPO training, and apply SFT on the rest most "difficult" and ambiguous data.

**Strengths:**

This paper addresses how to better utilize ambiguous preference data for LLM alignment. With prior works mostly discarding such data, this paper provides a simple yet effective way to leverage them.

Moreover, the proposed method is empirically validated on multiple benchmarks to improve over vanilla DPO, verifying the effectiveness of curriculum learning and SFT on ambiguous data.

**Weaknesses:**

- **Unclear motivation**: The authors connect the ambiguous preference pairs with the likelihood displacement problem, but do not provide solid evidence that the two are indeed related. This makes the motivation of adding an SFT loss on ambiguous data somewhat weak.
- **Limited novelty**: The sorted curriculum learning strategy and SFT loss are both straightforward and have been explored in prior works. The difference here is mainly the application to ambiguous preference data.
- **Weak Performance on Arena-Hard**: The proposed method appears to be weaker than other DPO variants on the Arena-Hard benchmark, which is a key benchmark for evaluating the model's ability.

**Questions:**

1. The "difficult" data are defined as those preference pairs with small differences in predicted scores.
However, these pairs are also likely to be (nearly) identical noisy samples, so the word "difficult" is somewhat misleading.
It would be helpful if the authors made this clearer in the paper.
2. Some settings of the ablation study in Figure 4 are missing. It does not include DPO + sort + discard difficult (i.e., MixDPO without SFT on difficult data) and IPO+sort, IPO+discard difficult. Can the authors include these experiments for a more complete comparison?

---

### Official Review · Reviewer_XjS6 · 2025-10-29

**Soundness:** 2
**Presentation:** 2
**Contribution:** 1
**Rating:** 2
**Confidence:** 4

**Summary:**

The authors analyse DPO-like methods make two observations:
1. Training LLMs on easy to difficult pairs (curriculum-style) helps performance
2. Using noisy / difficult pairs for SFT rather than discarding them helps performance

**Strengths:**

- The core idea is simple, intuitive, and easy to implement
- Strong reported results (especially on Length Control): The paper provides compelling evidence that its SFT phase on difficult pairs helps to counteract the DPO tendency towards reward hacking (e.g. through style changes like length)

**Weaknesses:**

The paper's two primary contributions are both of questionable novelty:
- easy-to-difficult curriculum benefitting alignment is not a new insight. As [1, 2, 3] and other recent works (e.g., Gao et al., 2025 the authors already cited) have already demonstrated curriculum-based data ordering is a known to improve preference optimization.

- Using SFT on difficult pairs rather than discarding them is also not entirely new. The authors frame their work as an alternative to methods that discard difficult data (e.g., SelectiveDPO). However, the idea of applying SFT (on the winning response) to "difficult" or "noisy" pairs, rather than a preference-based objective, has been explored by others [4].

Given that both core components (curriculum learning and SFT on noisy/difficult data) have been previously explored, the paper's main contribution is the specific combination of these two ideas. Even though the reported empirical results are strong, the insights feel more like an effective heuristic or an engineering-style combination of known techniques rather than a fundamental new insight for preference alignment.
Additionally, a few minor issues:
- lines 112 and 113 RLHF and DPO acronyms are redefined
- lines 130-132 SFT acronym is redefined
- line 154 LLM is redefined
- lines 156-157 DPO is redefined
- lines 278-279 I would also cite [5]

[1] Curry-DPO: Enhancing Alignment using Curriculum Learning & Ranked Preferences, https://arxiv.org/abs/2403.07230

[2] Towards Understanding Valuable Preference Data for Large Language Model Alignment, https://arxiv.org/abs/2510.13212

[3] Curriculum Direct Preference Optimization for Diffusion and Consistency Models, https://arxiv.org/abs/2405.13637

[4] Meta-Learning Objectives for Preference Optimization, https://arxiv.org/abs/2411.06568

[5] Is dpo superior to ppo for llm alignment? a comprehensive study, https://arxiv.org/abs/2404.10719

**Questions:**

I'd like the authors to ablate over doing SFT on both the rejected and accepted datapoint vs only doing SFT on the accepted datapoint.
i.e.:
Could the authors please provide an ablation that compares the current method against:
(a) The current method (SFT on $y_w$ only).
(b) Applying SFT on both the chosen ($y_w$) and rejected ($y_l$) responses. (This might be valuable if both responses are high-quality, which is likely for a low-margin pair).
(c) Applying SFT on $y_w$ and a form of "unlearning" (e.g., a negative log-likelihood loss) on $y_l$.

---

### Official Review · Reviewer_fgTm · 2025-10-30

**Soundness:** 2
**Presentation:** 2
**Contribution:** 2
**Rating:** 2
**Confidence:** 3

**Summary:**

This paper proposes MixDPO, a simple two-stage approach for preference-based alignment of large language models. The method applies Direct Preference Optimization (DPO) to easy, high-margin preference pairs and supervised fine-tuning (SFT) to difficult, low-margin pairs that are usually discarded in prior work. In the SFT stage, only the chosen responses from difficult pairs are used as training targets. Experiments on UltraFeedback and Argilla-7k datasets with LLaMA-3-8B, Mistral-7B, and Qwen-2.5-7B show modest but consistent improvements over several DPO variants on benchmarks such as AlpacaEval 2, Arena-Hard, and MT-Bench. Ablation studies further analyze threshold sensitivity, alternative difficulty metrics, and compatibility with other DPO variants.

**Strengths:**

- Addresses a practical and relatively underexplored aspect of preference learning: how to effectively utilize ambiguous preference pairs, offering a data-centric perspective.
- Proposes a simple and computationally lightweight modification that can be easily implemented on top of existing DPO pipelines.
- Reports empirical results across multiple base models and datasets, showing consistent, though modest, performance improvements.

**Weaknesses:**

The proposed method is rather ad-hoc, and the empirical evidence is not fully convincing. A deeper analysis, theoretical justification, or more comprehensive experiments would help evaluate its effectiveness.

- The method enforces a fixed two-stage schedule (DPO on easy → SFT on difficult), but does not test alternative schedules (e.g., SFT→DPO, or all-pairs DPO followed by SFT-on-all-chosen, and vice versa). Hence, the necessity of the proposed ordering remains unclear.

- The SFT term in MixDPO is applied only to chosen responses. Given that some rejected responses may have high absolute ratings (and some chosen responses may be low), a rating-thresholded SFT that ignores the role (chosen/rejected) could serve as a natural baseline. This variant is not evaluated, and including such straightforward variants would further support the validity of the proposed idea.

- The empirical results are presented without error bars or variance estimates, and it is unclear whether they are averaged over multiple runs. Given that improvements are modest (typically within 3–6%), it is difficult to assess whether the reported gains are statistically significant or reproducible.

- While the paper empirically shows that applying SFT to difficult pairs alleviates the degradation caused by likelihood displacement, it seems not to provide sufficient quantitative or theoretical analysis to explain why SFT mitigates this effect.

**Questions:**

- Could the authors clarify the definitions of each configuration in Figure 4? For example, “Ours” appears to correspond to DPO on easy pairs followed by SFT on chosen responses of difficult pairs. Does “Ours + discard difficult pairs” mean only DPO on easy pairs (sorted)? If so, is “DPO + sorted data” simply DPO with curriculum-style ordering instead of random order?

- Do the figures referred to as Figure 5a and Figure 5b in Section 6.2 correspond to Figure 4a and Figure 4b, respectively?

- The paper mentions a two-stage training paradigm (DPO → SFT) and reports multiple epochs in Table 7. Could the authors clarify whether DPO and SFT are alternated per epoch (e.g., DPO → SFT → DPO → SFT) or whether each stage is repeated for several epochs (e.g., DPO → DPO → SFT → SFT)? The current description leaves this aspect somewhat ambiguous.

- More comprehensive ablations that cover these design variants would help confirm the robustness of the method.

---

### Official Review · Reviewer_JnKu · 2025-10-31

**Soundness:** 2
**Presentation:** 2
**Contribution:** 2
**Rating:** 4
**Confidence:** 4

**Summary:**

This paper addresses an inefficiency in current PO : the handling of low reward margin examples. While prior work often discards these pairs to avoid noise, this paper argues they still contain valuable signal if optimized with the correct objective. The authors propose MixDPO, a method that sorts preference pairs by their rating margin. High-margin ("easy") pairs are trained with standard Direct Preference Optimization (DPO), while low-margin ("difficult") pairs are trained using only a Supervised Fine-Tuning (SFT) loss on the preferred response.

The authors though are ignoring some key baselines and are comparing with DPO and Simpo which are nearly 2 years old now. I encourage the authors to look at more recent works from 2024-25 where some of the concerns they are raising have been addressed.

**Strengths:**

Easy and Pragmatic Method:
The idea that data considered too noisy for contrastive learning might still be useful for strictly positive supervised learning is intuitively sound. Instead of filtering, the authors provide a way to utilize expensive preference annotations that would otherwise be wasted. (but note that some current methods do not need to do such filtering -- see more in weaknesses).

Important Problem Statement:
The likelihood displacement problem is one of the most fundamental failure modes of DPO. Fixing this is necessary to increase its adoption. The paper proposes threshold based SFT to solve this problem (but SFT has already been considered in literature) and this is well-justified, as SFT provides a purely positive learning signal without the displacement.

**Weaknesses:**

### **Novelty**
The novelty is relatively thin when placed in the context of prior work that already identified SFT as a fix for DPO instability. For example, Pal et al. [1] previously proposed DPO-Positive, explicitly adding SFT terms to mitigate failure modes like likelihood displacement. Given this, MixDPO is effectively a hyperparameter-guided switch for this known fix, which is a minor novelty.

### **Old baselines**
While MixDPO improves over vanilla DPO (achieving ~36% raw Win Rate and ~14% Length-Controlled Win Rate on AlpacaEval 2 with Llama-3-8B), recent work from late 2024 and 2025 has higher performance on the same model class. Does the conditional SFT improve these methods as well? See [1-5].


### **SIMPO Numbers**
The authors quote simpo's number as 24%, whereas the paper reports 44% on the same model (see Table 1 in the paper)? Am I missing something or are the numbers totally off?


#### **Hyperparameter Brittleness**
The method introduces a critical sensitivity to the difficulty threshold $\tau$. As shown in Table 3, selecting a suboptimal threshold (e.g., $\tau=1.5$) can degrade performance to below that of vanilla DPO. Can you show the hyperparameter ablation to find tau?.

---

### References

[1] Chen, H., He, G., Yuan, L., Cui, G., Su, H., & Zhu, J. (2024). Noise contrastive alignment of language models with explicit rewards. Advances in Neural Information Processing Systems, 37, 117784-117812.

[2] Wu, Y., Sun, Z., Hughes, R., Ji, K., Yang, Y., & Gu, Q. (2025). Self-play preference optimization for language model alignment., International Conference on Representation Learning (Vol. 2025, pp. 91558–91582).

[3] Gupta, T., et al. (2025). AMPO: Active Multi Preference Optimization for Self-play Preference Selection. ICML 2025.

[4] Gupta, T., et al. (2025). REFA: Reference Free Alignment with Fine-Grained Length Control. COLM 2025.

[5] Tang, X., et al. (2025). Game-Theoretic Regularized Self-Play Alignment of Large Language Models. arXiv preprint arXiv:2503.00030.

**Questions:**

1. SFT memorizes data, and hence there is a chance that performance on related benchmarks drops. Could you do a quick check on some benchmarks in math/other non instruction following datasets?

2. Could you provide a comparison against a uniform mixed-loss baseline, where you train on all data with $L = L_{DPO} + \lambda L_{SFT}$? Please do this for different values of lambda. This comparison is useful to establish whether your explicit sorting and thresholding mechanism provides value beyond the SFT regularization.

3. Is the staged "curriculum" aspect strictly necessary for performance? Pls consider ablations where MixDPO is applied in a single stage with random batches.

---

### Note · Authors · 2025-11-17

I have read and agree with the venue's withdrawal policy on behalf of myself and my co-authors.